# Analysis of the Influencing Factors on S-Band Sea Spikes

**Peng Zhao** [1,2] , **Zhensen Wu** [1,*] , **Yushi Zhang** [2] **and Dong Zhu** [2]

1   School of Physics, Xidian University, Xi'an 710071, China
2   National Key Laboratory of Electromagnetic Environment, China Research Institute
    of Radiowave Propagation, Qingdao 266107, China
*   Correspondence: wuzhs@mail.xidian.edu.cn

**Abstract:** While detecting targets on the sea by radar that looks downward, sea spikes similar to the target echoes cause false alarms, especially at low grazing angles. To suppress the interference of sea spikes on target detection, it is important to study the factors influencing the occurrence probability of sea spikes during radar inspections. Based on measured sea clutter data, this study uses the amplitude threshold, duration time, and interval time to identify sea spikes from sea clutter and uses the pulse number per unit time to characterize their occurrence probability. The influences of the grazing angle, wave height, wave direction, and wind speed on the occurrence probability of sea spikes were analyzed. The results indicate that the occurrence probability of sea spikes increases exponentially with decreasing grazing angle in the range from $0.7°$ to $7.1°$, and linearly with increasing wind speed. Wave direction has little or no influence on the probability of sea spikes. At the smaller grazing angles, from $0.7°$ to $1.7°$, the influence of wave height on the probability of sea spikes' occurrence is obvious, showing a linear trend, but in the range of $2.6°$ to $7.1°$, the influence is not obvious. In addition, the occurrence probability of sea spikes is found greater at wave heights from 0.9 m to 1.1 m relative to other wave heights, which is worthy of further study.

**Keywords:** sea spikes; low grazing angles; occurrence probability; wave heights

## 1. Introduction

Sea spikes are strong and discrete returns from the sea, often similar to targets, which seriously affect the performance of radars in the detection and tracking of sea targets with low radar cross sections (RCS) [1]. Compared to common sea clutter, sea spikes often lead to impulse characteristics in the time domain and waveforms are characterized by long trails in the amplitude distribution. They present a large Doppler component with a wide bandwidth and strong backscatter power. In addition, they exhibit a horizontal transmit and horizontal receive (HH) return, which is equal to or greater than the vertical transmit and vertical receive (VV) return [2]. Sea spikes have been found to cause false target detections, which has fueled an urgent need to study their characteristics under different sea conditions to effectively suppress clutter and improve the radar detection performance [3–7].

The research on sea spikes mainly focuses on two important aspects: (1) the separation of sea spikes from general sea clutter, that is, the method of judging and extracting sea spikes, and (2) the influence of radar and marine environmental parameters on the occurrence probability of sea spikes [2]. In the present study, we focus on the latter problem, based on the analysis of measured sea clutter data at low grazing angles. Posner et al. [3,4] analyzed sea clutter data measured by a high-resolution X-band radar at low grazing angles. They illustrated that the sea spike phenomenon in the horizontal polarization and upwind direction is stronger than that in other situations. Melief et al. [5] analyzed three sets of high-resolution, coherent, and polarimetric radar sea clutter data. They revealed that spiking events possess higher power, polarization ratio, and velocity. Hwang et al. [6] indicated that the key results of the breaking effects significantly increased the

Doppler velocity of both polarizations, drastically enhancing the horizontally polarized backscattering cross-section. Greco et al. [7] established an identification method for sea radar clutter spikes by introducing three sea spike-defining parameters: spike amplitude, minimum spike width, and minimum interval between spikes. Guan et al. [8] studied an algebraic fractal model of sea spikes, and the results showed that this method has good detection efficiency in the case of a low signal-to-noise ratio. Chen et al. [9] proposed an algorithm based on a short-time Fourier transform for target extraction and detection in the sea-peak background, which demonstrated high detection efficiency. For sea spikes with characteristics similar to moving targets, an improved target detection algorithm under the background of sea spikes was reported by Song [10]. Other researchers [11–13] have investigated the occurrence mechanism of sea spikes, which provides a lot of theoretical support for the judgment basis.

Research on the formation of sea spikes is mainly based on the analysis of experimental data [14,15]. Smith et al. [16] inferred that sea spikes occurred more frequently when there were breaking waves in the radar illuminating area. Gutnik et al. [17] discovered that the probability of sea spikes' occurrence in a single distance gate, at a certain time, can be approximately characterized by a Poisson distribution. Yang et al. [18] proposed a statistical characteristic method for sea spikes from the spatial and time domains. Xie et al. [19] researched the relationship between sea spikes and time-frequency distribution in a single range resolution unit of IPIX clutter data for sea spike effects. Based on measured sea clutter data, Huang et al. [20] reported the characteristics of amplitude, time correlation, Doppler spectrum, and fractional power spectrum of sea spikes under different sea conditions and polarization modes. Gu [21] proposed a sea spike suppression method to detect small floating targets based on multi-features and principal component analysis. Zhao et al. [22] proposed a sea spike suppression method based on an optimum polarization ratio in airborne SAR Images. Yurovsky et al. [23] used the Ka-band radar to measure the sea spikes. Most of the aforementioned studies were carried out under conditions of high resolution, small grazing angles, and high sea states. However, it has been recently reported that sea spikes can also occur under low resolution conditions and middle or large grazing angles. Li et al. [24] studied the possibility of UHF and L-band sea spikes' formation. Their results showed that the amplitude of the HH polarization was significantly greater than that of the VV polarization in both the UHF and L-bands. This phenomenon was more pronounced in the L- rather than the UHF band. Rosenberg et al. [25] studied the sea clutter data of the Ingara radar and found that sea spikes also exist in the case of medium and large grazing angles.

The factors influencing the formation of sea spikes are polyphyletic. Therefore, the completeness of sea clutter data and matched environmental parameters are very important for statistical analysis. However, the volume of such information from the measured sea clutter data used in previous studies is insufficient, limiting the accuracy of analyses results. In this study, we analyze the influence of different factors on the probability of sea spikes' occurrence, based on the S-band sea clutter data, and the environmental parameters collected synchronously, measured by the China Radio Propagation Institute on Lingshan Island. The remainder of this paper is organized as follows. The S-band radar parameters and data acquisition are presented in Section 2. In Section 3, we introduce methods for distinguishing and screening sea spikes and we define their occurrence probability. In Section 4, the relationship between the formation of sea spikes and factors such as grazing angle, wind speed, and wave height are analyzed. Finally, conclusions are presented in Section 5.

## 2. Sea Clutter Data and Measured Parameters

Based on the S-band radar sea clutter data measured by the China Radio Propagation Research Institute on Lingshan Island in the Yellow Sea, this study analyzes the factors influencing sea spikes. Figure 1 shows the installation location of the radar on Lingshan Island and measured areas of the sea surface. The S-band radar is a shore-based radar with

a pulse Doppler system, HH polarization, and range resolution of 60 m. The experiments were conducted between October 2013 and September 2014. In 2013, the altitude of the radar was approximately 70 m, the azimuth was fixed at 110°, the main beam of the radar antenna was pitching down by 0.5°, and the range of the grazing angle was 0.7°–1.7°, corresponding to the radial distance from 2.3 km to 5.7 km, marked by white lines in Figure 1. In 2014, the radar was erected at an altitude of approximately 430 m. The main beam of the radar antenna was pitching down by 5°, the azimuth was fixed at 65°, and the grazing angle was 2.6°–7.1°, corresponding to the radial distance from 3.5 km to 9.5 km, marked by pink lines in Figure 1.

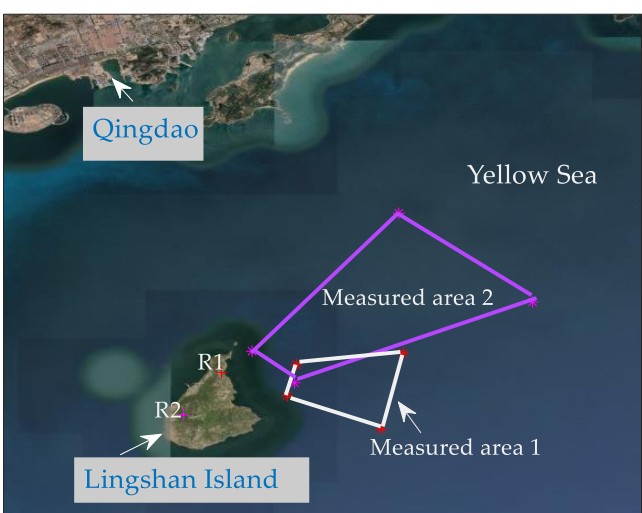

**Figure 1.** Radar location and measurement region.

The measurement duration for a single group of sea clutter data was approximately 120 s. The daily experiment was carried out from 6 am to 10 pm, traversing a variety of wave and meteorological conditions. During the acquisition of sea clutter data, wave measurement buoys and ultrasonic meteorological observation systems were used synchronously to obtain oceanic and meteorological parameters of the measurement region, such as wave height, wave direction, wind speed, and wind direction. The measurement equipment is shown in Figure 2.

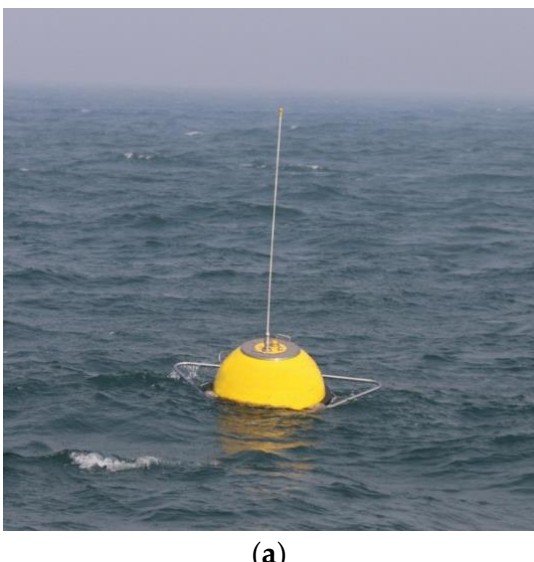

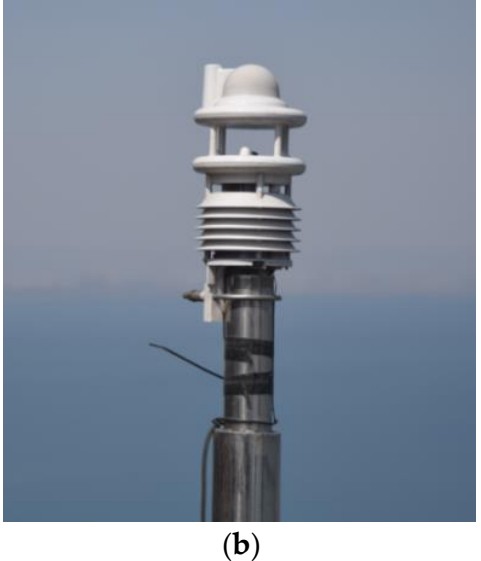

**(a)**        **(b)**

**Figure 2.** Measurement equipment: (**a**) buoy, (**b**) ultrasonic meteorological observing system.

As shown in Table 1, 2317 sets of sea clutter data were obtained during the experiment, including 466 sets of data collected in 2013 and 1851 sets of data collected in 2014. The wave height ranges from 0.1 m to 2.0 m, and the relative wave direction includes up-wave, cross-wave and oblique-cross wave, according to the relative angles between the radar beam direction and swell direction of sea. The wind speed ranges from 0.1 m/s to 14 m/s. Figure 3 shows the variation in oceanic parameters with dates.

**Table 1.** Sea clutter data and oceanic parameters.

| Test Time | 2013 | 2014 |
|---|---|---|
| Total data | 466 | 1851 |
| Grazing angle $\theta$ (°) | 0.7~1. 7 | 2.6~7.1 |
| Wind speed $w$ (m/s) | 0.9~13.2 | 0.1~14 |
| Wave height $h$ (m) | 0.1~1.4 | 0.1~2.0 |
| Wave direction $\Delta\varphi$ (°) | up wave ($\|\Delta\varphi\| < 20°$)<br>cross wave ($80° < \|\Delta\varphi\| < 100°$)<br>oblique-cross wave ($20° < \|\Delta\varphi\| < 80°$ and $100° < \|\Delta\varphi\| < 160°$)<br>up wave ($\|\Delta\varphi\| < 20°$) | |
| Test duration (s) | 120 | |

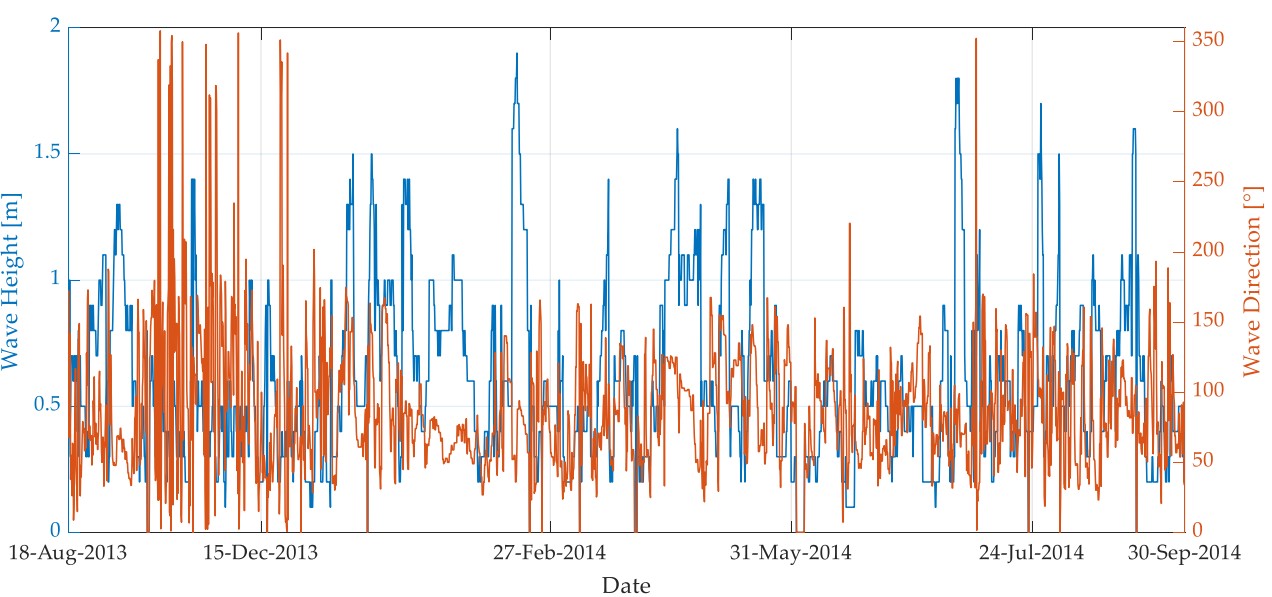

**Figure 3.** Variation of the oceanic parameters during the entire test period.

## 3. Discrimination and Statistical Methods of Sea Spikes

### 3.1. Discrimination Method of Sea Spikes

To effectively analyze the influence of different factors on the formation of sea spikes, it is primarily necessary to separate the sea spikes from the clutter background. Posner [3] analyzed the average duration and interval of sea spikes based on measured data. He illustrated that the spike amplitude threshold, minimum spike width, and minimum spike interval are three important parameters for describing sea spikes. The sea spike in the IPIX data was well distinguished from the time domain by Greco et al. based on this method [7]. Owing to the simplicity and convenience of the discrimination method, it was later used by more scholars [8,12,21], in spite of some associated limitations. The discrimination rules and parameters are defined as follows:

1. Spike amplitude threshold: If sea spikes are formed, the radar echo signal must be greater than the spike amplitude threshold. The calculation formula for the spike amplitude threshold is generally expressed as:

$$\lambda = \sqrt{\frac{\alpha}{N_c N_t} \sum_{i=1}^{N_c} \sum_{j=1}^{N_t} |Z_i(j)|^2},$$ (1)

where $\lambda$ is the spike amplitude threshold intensity, $\alpha$ is the threshold coefficient, $N_c$ is the number of pulses, $N_t$ is the number of range cells, and $|Z_i(j)|$ represents the amplitude of the sea echoes.

2. Minimum spike width: The amplitudes of the returns must remain above the spike amplitude threshold for at least the specified minimum spike width.

3. Minimum spike interval: If the amplitudes of the returns fall below the spike amplitude threshold, they may not remain below the spike amplitude threshold for longer than the specified minimum interval between spikes. If they remain below longer, then this set of returns would be considered to consist of separate sea spike candidates.

According to the above conditions, the sea spike can be separated from the clutter background. It should be noted that these three parameters may differ in different radars and environments. The parameters adopted in this study were as follows: the minimum spike width was 100 ms, the minimum spike interval was 500 ms, and the spike threshold was five times the average power of sea clutter, i.e., $\alpha = 5$. Figure 4a shows the range-time image of a group of sea clutter data, and Figure 4b shows the sea spikes' discrimination results at 80th rang bin in Figure 4a. The red line represents the clutter data identified as sea spikes, the blue line represents the sea clutter background data, and the black dotted line represents the set spike amplitude threshold.

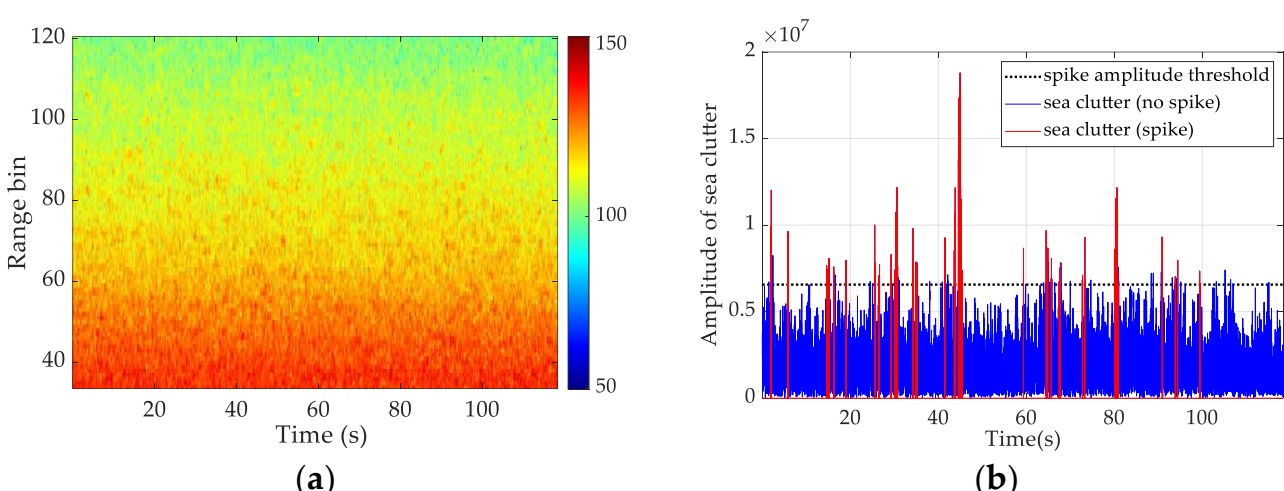

**Figure 4.** Example of sea spike determination. (**a**) Range-time image of sea clutter data; (**b**) Sea spike determination results at 80th rang bin.

### 3.2. Statistical Method

In this study, the ratio of the number of sea spike pulses per unit time was used as the probability of sea spike occurrence using the following formula:

$$r = \frac{N_s}{N_t} \times 100\%,$$ (2)

where $r$ is the occurrence probability of sea spikes, $N_t$ is the number of pulses of a single range cell, and $N_s$ is the number of pulses occupied by sea spikes.

As shown in Figure 4a, each group of data presents the matrix form of the distance gate pulse number. Here we adopt the method of sliding determination of every three distance gates to carry out sea spike determination individually and calculate the sea spike occurrence probability of each range cell according to (2).

When analyzing the factors influencing sea spikes, the method of statistical comparison of mean values under different conditions was adopted. In terms of the change in the distance gate, the probability of sea spikes at different distance gates in the entire dataset is averaged. Assume that $D$ represents a clutter dataset that contains $M$ groups of clutter data; that is,

$$D = [D_1 \ D_2 \ \cdots \ D_M] \tag{3}$$

Let $\overline{D}_r(n)$ be the mean value of the sea peak occurrence probability of the $n$-th range cell for all data in the dataset.

$$\overline{D}_r(n) = \frac{1}{M} \sum_{m=1}^{M} r_m(n) \tag{4}$$

where $r_m(n)$ represents the occurrence probability of sea spikes in the $n$-th range cell of the data of group $m$, with $n = 1, 2, \ldots N_c$. From (4), the mean value of the sea spike occurrence probability in different range cells of the entire dataset can be obtained by:

$$\overline{D} = [\overline{D}_r(1)\overline{D}_r(2)\cdots\overline{D}_r(N_c)] \tag{5}$$

## 4. Influencing Factors of Sea Spikes

According to the generation mechanism of sea spikes, the sea surface changes from a steady to an unsteady state, causing the emergence of sharp crests and breaking waves on the sea surface, resulting in an enhancement effect of backscattering. Therefore, the grazing angles of the radar and sea conditions were strongly related to the generation of sea spikes. Based on the measured sea clutter data, this section conducts statistical research from different aspects, such as grazing angles, wind speed, wave height, and wave direction, and analyzes the impact of different factors on the occurrence of sea spikes.

### 4.1. Grazing Angle

A geometric relationship exists between grazing angles and range gates. Thus, in the following, the range gates were converted into grazing angles. Because the elevations of the radar are inconsistent in 2013 and 2014, the data measured in these two periods correspond to different grazing angle ranges. The range of the grazing angle was 0.7°–1.7° in 2013 and 2.6°–7.1° in 2014.

According to the Formulas (3)–(5), we have obtained sea spikes' probability at every range gate for all datasets. Figure 5 shows the variation trend of the sea spikes' probability with the grazing angle in the three groups of data at different grazing angle ranges. Figure 5a,b presents datasets in 2014, and the grazing angle ranges are 2.6° to 7.5° and 3.7° to 7.5° separately. Figure 5c is a dataset in 2013 with grazing angle from 1° to 1.7°. As shown in Figure 5, the probability of sea spikes increases gradually with a decrease in grazing angle. This result implies that phenomenon of sea spikes is more likely to occur at smaller grazing angles, which is consistent with conclusions in the literature [16,17,23]. At the same time, we perform curve fitting for these three groups of data and find that they almost show a linear trend of change at their ranges of grazing angle.

To analyze the trend of the probability of sea spike occurrence with grazing angles more accurately, the probability of sea spike occurrence for all data in 2013 and 2014 was statistically averaged according to different grazing angles, and the results are shown in Figure 6. The average probability of sea spikes is found to increase with a decrease in grazing angle in both 2013 and 2014 datasets. It is worth noting that the grazing angle in 2013 was smaller and the overall probability of sea spikes was greater than that in 2014. This phenomenon is consistent with that shown in Figure 5. As mentioned in [2,3], the sea

spike is correlated to structures of sea waves and the direction of radar observation. Thus, it is reasonable that there is a bigger probability of sea spikes at lower grazing angles.

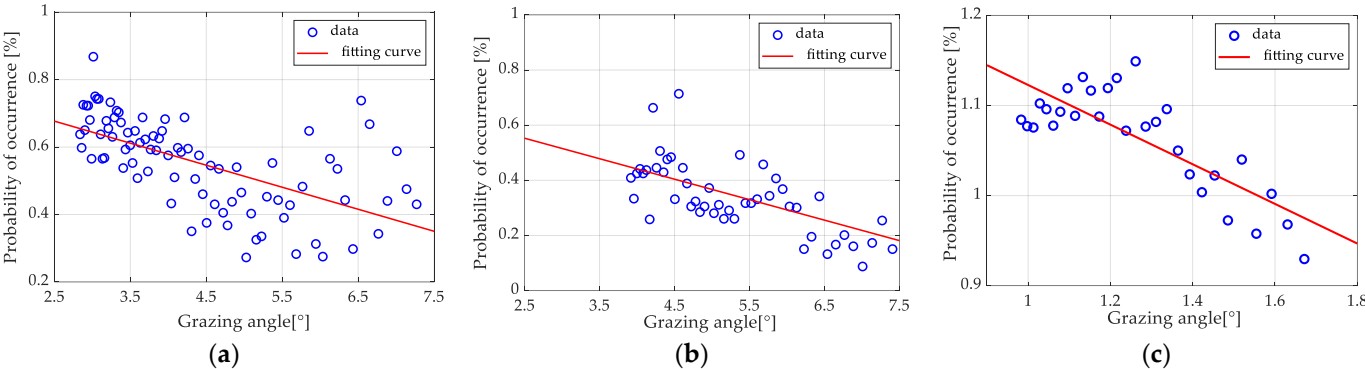

**Figure 5.** Probability of sea spikes with grazing angle of three datasets at different grazing angle ranges. (**a**) Range of grazing angle, 2.6°–7.5°. (**b**) Range of grazing angle, 3.7°–7.5°. (**c**) Range of grazing angle, 1°–1.7°.

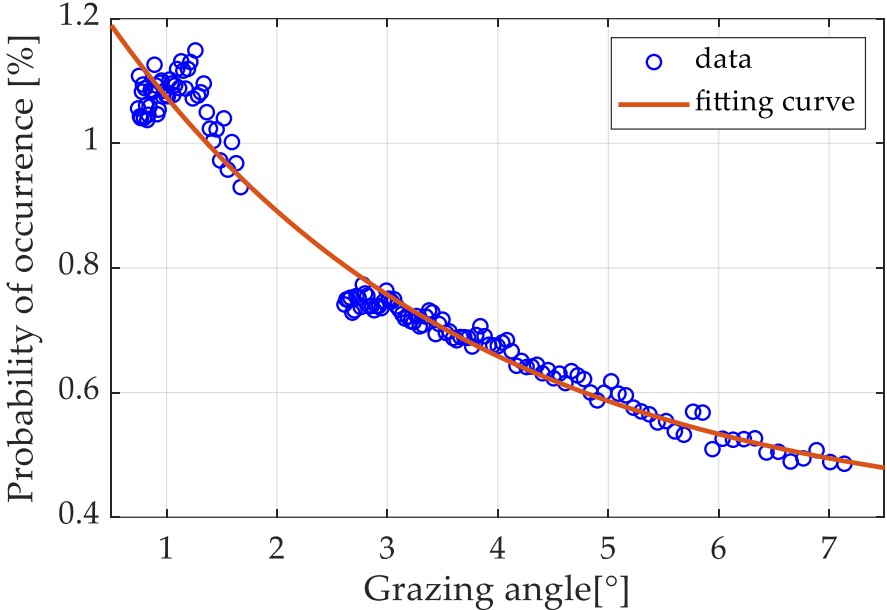

**Figure 6.** Probability of sea spikes' occurrence with grazing angles among all data.

Furthermore, through regression fitting of the data, it was found that the probability of a sea spike increases exponentially with grazing angle decreasing at the range from 0.7° to 7.1°, and the fitting formula is as follows:

$$\overline{D}_r(\theta) = 0.94e^{-0.31\theta} + 0.39,\tag{6}$$

where $\theta$ represents the grazing angle.

Formula (6) seems to be contrary to the result of Figure 5, but from the relative relationship between the entirety and locality, it can be explained. When the grazing angle range is not big enough, as shown in Figure 5, the data can be both fitted by exponential form and linear form, and it is hard to discriminate. However, when the two ranges of a grazing angle are combined, the exponential form is more fitted to the averaged values. Therefore, from the point of the whole trend for the range from 0.7° to 7.1°, it could be supposed to the exponential form. In addition, there is a lack in our data from 1.7° to 2.6°, so the accuracy of Formula (6) needs to be further verified.

### 4.2. Oceanic Parameters

Wave heights, wave direction, and wind speeds are important parameters for characterizing changes in the sea surface. They are also important factors affecting the characteristics of sea clutter. To obtain the impact of oceanic parameters on the formation of sea spikes, we analyze all sea clutter data, and then make statistical analysis on the sea spike occurrence probability for different wave directions, wave heights, and wind speeds. According to the variation of average values with wave directions, wave heights, and wind speeds, we infer the impact relationship of oceanic parameters.

Figure 7 shows the variation in the mean probability of sea spikes for different wave heights in different wave directions. From the comparison of the three curves in Figure 7, there are no obvious rules for the entire trend, although at some wave height ranges, the values of different wave directions have a certain size order. Thus, it could be inferred that wave directions have little or no influence on the probability of sea spikes.

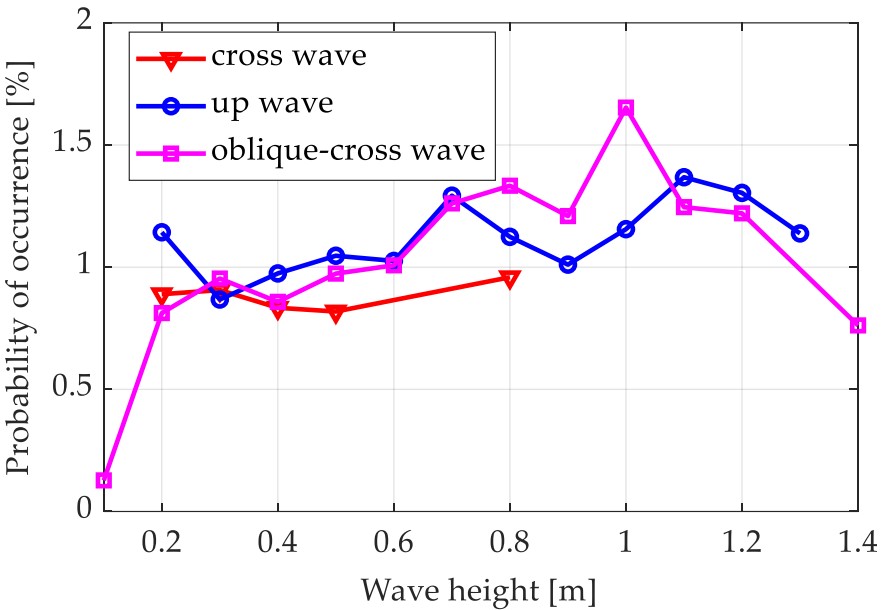

**Figure 7.** Comparison of the occurrence probability of sea spikes in a different wave direction.

It should be noted that in Figure 7, there is some relationship between wave heights and the probability of sea spikes. In the up wave direction, the probability value of sea spikes increases slowly with the wave height. However, in the oblique-cross wave direction, the probability increases from 0.1 m to 1 m and decreases from 1 m to 1.4 m, and especially for wave heights approaching 1 m, it is significantly higher with a steeper rate of change.

From the above analysis, we know that grazing angles are a major factor for sea spike occurrence probability and wave direction is an unimportant factor. Thus, in order to obtain the relationship between wave heights and the probability of sea spikes' occurrence, we separately analyze the sea clutter data in 2013 and 2014 at a fixed grazing angle. Figure 8 shows the statistical analysis results. Figure 8a shows the data in 2013, corresponding to the 0.8° grazing angle, and Figure 8b shows the data in 2014, corresponding to the 5.1° grazing angle. The blue circles represent the distribution of the sea spike probability of multiple groups of data at different wave heights and the probability value at the same wave height also shows a certain dynamic range. The red curves correspond to the average probability value at each wave height, and the pink straight line is a linear fitting curve.

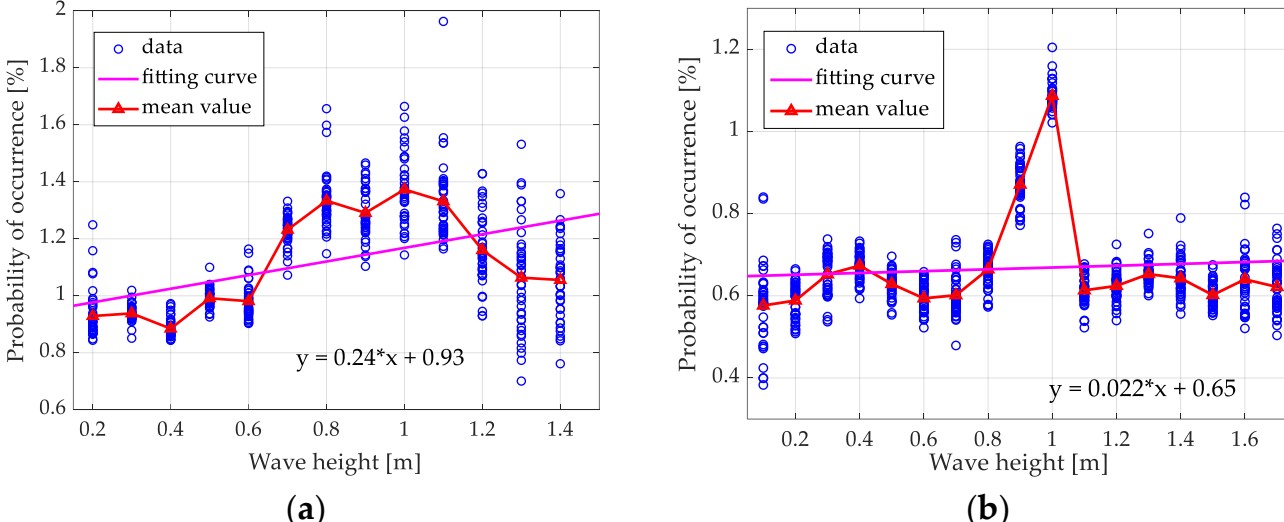

**Figure 8.** Variation of sea spike occurrence probability with wave height. (**a**) 2013, at 0.8° grazing angle. (**b**) 2014, at 5.1° grazing angle.

As shown in Figure 8a, the dynamic range of probability value at each wave height is among 0.2% to 0.8%, which is hard to summarize the variation rules. However, when observing the average values at each wave height, the probability increases from 0.2 m to 1 m and decreases from 1 m to 1.4 m. For the linear fitting curve, it shows that the probability of sea spikes at 0.8° grazing angle gradually increased with increasing wave height, and the slope of the linear is 0.24. In contrast, in Figure 8b, there is no obvious trend with wave heights increasing for the data at 5.1° grazing angle. If not the probability values at 0.9 m and 1 m are much bigger than the other wave heights, it can almost be considered that the wave height has no influence on the probability value.

From the comparison of Figure 8a,b, it can be suggested that the influence of the wave height on the probability of a sea spike is more significant at smaller grazing angles. The fitting formula for wave height at 0.8° grazing angle is as follows:

$$\overline{D}_r(h) = 0.24h + 0.93, \quad (\theta = 1.7°), \tag{7}$$

where *h* refers to the wave heights.

Relative to the influence of wave heights on the trend of sea spike probability, the higher probability at 1 m wave height is more noticeable. Whether this is a common phenomenon or a special phenomenon for only the S-band needs further studies.

In addition to analyzing the influence of the wave height and wave direction, the influence of the wind speed was further investigated. In common with the analysis of wave height, we also separately analyze the sea clutter data in 2013 and 2014 at a fixed grazing angle. The average value at each wind speed has been calculated for the data at 1.5° grazing angle and 6° grazing angle. Figure 9 shows the relationship between the probability of sea spike occurrence and the wind speed.

It can be seen that with an increase in the wind speed, the probability of a sea spike gradually increases. We performed curve fitting for the two datasets and found that they almost show a linear trend of change with wind speed. The linear fitting formulas are as follows:

$$\overline{D}_r(\omega) = 0.032\omega + 0.57, \quad (\theta = 1.5°), \tag{8}$$

$$\overline{D}_r(\omega) = 0.04\omega + 0.4, \quad (\theta = 6°), \tag{9}$$

where $\omega$ refers to the wind speeds.

Although Formulas (8) and (9) are only fitting results under two grazing angles, they have similar forms and coefficient values, so it could be inferred that the linear trend by wind speed for data at most other grazing angles is also applicable.

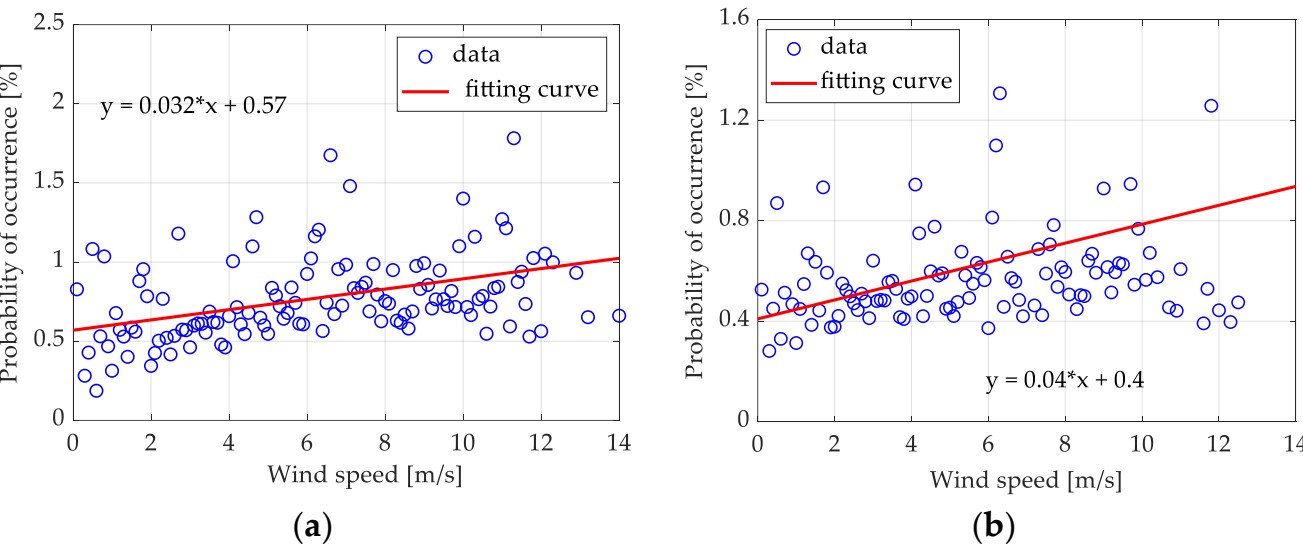

**Figure 9.** Variation of sea spike probability with wind speed. (**a**) 2013, at 1.5° grazing angle. (**b**) 2014, at 6° grazing angle.

## 5. Conclusions

Based on S-band measured sea clutter data, the probability of sea spike formation was investigated in this study. Using the identification methods of sea spikes, the influence of grazing angles, wave heights, wave directions, and wind speeds on the probability of sea spikes' occurrence were analyzed. The main conclusions are as follows:

The probability of sea spikes' occurrence decreases exponentially with an increase in grazing angles; that is, sea spikes are more likely to occur at low grazing angles. Moreover, the exponential form of trend with grazing angles is among the observed range of this study, from 0.7° to 7.1°. Whether this result is suitable for other grazing angles requires further verification of measured data.

At the smaller grazing angles, from 0.7° to 1.7°, the influence of wave height on the probability of sea spikes' occurrence is obvious, showing a linear trend. In the range of 2.6° to 7.1°, the influence of wave height is not obvious. However, it is worth nothing that when the wave height is about 1 m, the probability of sea spikes' occurrence is significantly higher than other wave heights. Further studies are required to determine whether this phenomenon is unique to the S-band.

For the influence of wind speed, the probability of sea spikes shows a linear increasing trend with an increase in wind speed. In addition, the wave direction has little or no impact on the probability of sea spikes.

This paper gives certain beneficial rules between the occurrence of sea spikes and the ocean conditions though data analysis and it is helpful for understanding and mastering the characteristics of sea spikes at low grazing angles for low resolution radars. Our future work will further verify the accuracy of Formulas (6)–(9) with more analysis of measured data, and we will further promote the research on higher wave bands, such as on the X-band or Ku-band.

**Author Contributions:** Conceptualization, Z.W.; methodology, Y.Z.; formal analysis, P.Z. and D.Z.; investigation, P.Z.; resources, Y.Z.; data curation, P.Z.; writing—original draft preparation, P.Z.; writing—review and editing, P.Z.; visualization, D.Z.; supervision, Y.Z.; project administration, Z.W.; funding acquisition, P.Z. All authors have read and agreed to the published version of the manuscript.

**Funding:** This research was funded by the National Natural Science Foundation of China, under Grant U2006207.

**Data Availability Statement:** The data are not publicly available due to privacy.

**Acknowledgments:** We would like to thank our colleagues for their contributions during the experiments, and data pretreatment results in support of the manuscript.

**Conflicts of Interest:** The authors declare no conflict of interest.

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
