# Peer review of "Analysis of the Influencing Factors on S-Band Sea Spikes"

_electronics, doi:10.3390/electronics11244225_

Round 1

Reviewer 1 Report

Abstract: "The results indicate that the occurrence probability of sea spikes increases exponentially with decreasing grazing angle, and linearly with increasing wave height and wind speed." There is a lack of justification within the main part of the manuscript.

Figure 1: What is the impact of land surface topography in the second experiment?

Section 4.1. The results must be analyzed. There is a lack of physical interpretation.

Section 4.2. This section must be improved. It's hard to extract definitive conclusions from these figures.

Conclusions: This is a research article. This is not a technical note. Please, elaborate more.

The manuscript must improve significantly, and resubmit as a letter. If the authors decide to resubmit as an article, please, expand the research goals. 

Author Response

Dear Editors and Reviewers:

Thank you for your letter and for the reviewers’ comments concerning our manuscript entitled “Analysis of the influencing factors on S-band sea spikes” (ID: electronics-1965125). Those comments are all valuable and very helpful to revise and improve our manuscript, as well as the important guiding significance to our researches. We have studied comments carefully and have made correction which we hope meet with approval.

Reviewer 2 Report

1. What is the main question addressed by the research?

Due to the fact that sea spikes can alter the radar results, a detailed analysis is carried out in the paper to find the factors that influence sea spikes.

2. Do you consider the topic original or relevant in the field? Does it address a specific gap in the field?

The topic is relevant and as I know the surppression of sea spikes from the radar detection was carried out only in some simplified circumstances (not very large waves) [Research on the Sea Spike Suppression Based on Range Domain Characteristics of Relatively High Resolution Radar, Sun Hanqi, Wan Xiaoyong doi:10.1088/1742-6596/1607/1/012077].

3. What does it add to the subject area compared with other published material?

The modeling of the sea spikes

4. What specific improvements should the authors consider regarding the methodology? What further controls should be considered?

The methodology is safe. In the future they should look also at modeling higher waves.

5. Are the conclusions consistent with the evidence and arguments presented and do they address the main question posed?

The conclusions are consistent with the provided analysis; maybe a larger data set would improve the accuracy of the model.

6. Are the references appropriate?

The references are appropriate.

7. Please include any additional comments on the tables and figures.

The tables and figures are clear. No corrections are required.

Author Response

Dear Editors and Reviewers:

Thank you for your letter and for the reviewers’ comments concerning our manuscript entitled “Analysis of the influencing factors on S-band sea spikes” (ID: electronics-1965125). Those comments are all valuable and very helpful to revise and improve our manuscript, as well as the important guiding significance to our researches. We have studied comments carefully and have made correction which we hope meet with approval. Revised portion are marked in red in the paper.

Reviewer 3 Report

The influences of the grazing angle, wave height, wave direction, and wind speed on the occurrence probability of sea spikes were analyzed.

The results indicate that the occurrence probability of sea spikes increases exponentially with decreasing grazing angle and linearly with increasing wave height and wind speed.

The paper idea is good.

Some gaps between text. Please remove.

Novelty should be clearly stated.

Typo errors exist.

Images resolutions are not good.

Future works are not given.

Some references are old.

Author Response

(The authors gave the same response as above.)

Reviewer 4 Report

In this piece of work, the authors analyzed amplitude threshold, duration time, and interval time data from sea clutter radar inspection and discovered the correlation of sea spike probability to grazing angle, wave height, wave direction, and wind speed.

Compared to the previous study from other researchers, the authors acquired a larger quantity of sampling data, and the conclusion of sea spikes probability to grazing angles, wave height, and wind speed are clearly elaborated and provide meaningful insight for further study.

I do find that the introduction section is a little bit lengthy by stating each researcher's previous work. A shortened introduction cohesively summarizing relevant previous work will make the manuscript more concise. But I am ok with publishing in the current form. 

Author Response

(The authors gave the same response as above.)
